# Clinical Characteristics and Disability Progression of Early- and Late-Onset Multiple Sclerosis Compared to Adult-Onset Multiple Sclerosis

**DOI:** 10.3390/jcm9051326

**Published:** 2020-05-02

**Authors:** Omid Mirmosayyeb, Serge Brand, Mahdi Barzegar, Alireza Afshari-Safavi, Nasim Nehzat, Vahid Shaygannejad, Dena Sadeghi Bahmani

**Affiliations:** 1Isfahan Neurosciences Research Center, Isfahan University of Medical Sciences, Isfahan 81746-73461, Iran; omid.mirmosayyeb@gmail.com (O.M.); barzegar_mahdi73@yahoo.com (M.B.); N.nehzat96@gmail.com (N.N.); 2Universal Council of Epidemiology (UCE), Universal Scientific Education and Research Network (USERN), Tehran University of Medical Sciences, Tehran 14197-33151, Iran; 3Department of Neurology, Isfahan University of Medical Sciences, Isfahan 81746-73461, Iran; 4Center of Depression, Stress and Sleep Disorders, Psychiatric Clinics (UPK), University of Basel, 4002 Basel, Switzerland; serge.brand@upk.ch (S.B.); dena.sadeghibahmani@upk.ch (D.S.B.); 5Division of Sport Science and Psychosocial Health, Department of Sport, Exercise, and Health, University of Basel, 4032 Basel, Switzerland; 6Substance Abuse Prevention Research Center, Health Institute, Kermanshah University of Medical Sciences (KUMS), Kermanshah 6719851351, Iran; 7Sleep Disorders Research Center, Health Institute, Kermanshah University of Medical Sciences (KUMS), Kermanshah 6719851351, Iran; 8School of Medicine, Tehran University of Medical Sciences (TUMS), Tehran 1416753955, Iran; 9Department of Biostatistics and Epidemiology, Faculty of Health, North Khorasan University of Medical Sciences, Bojnurd 74877-94149, Iran; a.afshari@nkums.ac.ir; 10Faculty of Pharmacy, Ahvaz Jundishapur University of Medical Sciences, Ahvaz 6135715794, Iran; 11Departments of Physical Therapy, University of Alabama at Birmingham, Birmingham, AL 35209, USA

**Keywords:** multiple sclerosis, age of onset, early onset, late onset, magnetic resonance imaging, predictors, EDSS score, relapsing-remitting MS, secondary progressive MS

## Abstract

Background: Compared to the adult onset of multiple sclerosis (AOMS), both early-onset (EOMS) and late-onset (LOMS) are much less frequent, but are often under- or misdiagnosed. The aims of the present study were: 1. To compare demographic and clinical features of individuals with EOMS, AOMS and LOMS, and 2. To identify predictors for disability progression from relapsing remitting MS (RRMS) to secondary progressive MS (SPMS). Method: Data were taken from the Isfahan Hakim MS database. Cases were classified as EOMS (MS onset ≤18 years), LOMS (MS onset >50 years) and AOMS (MS >18 and ≤50 years). Patients’ demographic and clinical (initial symptoms; course of disease; disease patterns from MRI; disease progress) information were gathered and assessed. Kaplan–Meier and Cox proportional hazard regressions were conducted to determine differences between the three groups in the time lapse in conversion from relapsing remitting MS to secondary progressive MS. Results: A total of 2627 MS cases were assessed; of these 127 were EOMS, 84 LOMS and 2416 AOMS. The mean age of those with EOMS was 14.5 years; key symptoms were visual impairments, brain stem dysfunction, sensory disturbances and motor dysfunctions. On average, 24.6 years after disease onset, 14.2% with relapsing remitting MS (RRMS) were diagnosed with secondary progressive MS (SPMS). The key predictor variable was a higher Expanded Disability Status Scale (EDSS) score at disease onset. Compared to individuals with AOMS and LOMS, those with EOMS more often had one or two relapses in the first two years, and more often gadolinium-enhancing brain lesions. For individuals with AOMS, mean age was 29.4 years; key symptoms were sensory disturbances, motor dysfunctions and visual impairments. On average, 20.5 years after disease onset, 15.6% with RRMS progressed to SPMS. The key predictors at disease onset were: a higher EDSS score, younger age, a shorter inter-attack interval and spinal lesions. Compared to individuals with EOMS and LOMS, individuals with AOMS more often had either no or three and more relapses in the first two years. For individuals with LOMS, mean age was 53.8 years; key symptoms were motor dysfunctions, sensory disturbances and visual impairments. On average, 14 years after disease onset, 25.3% with RRMS switched to an SPMS. The key predictors at disease onset were: occurrence of spinal lesions and spinal gadolinium-enhancement. Compared to individuals with EOMS and AOMS, individuals with LOMS more often had no relapses in the first two years, and higher EDSS scores at disease onset and at follow-up. Conclusion: Among a large sample of MS sufferers, cases with early onset and late onset are observable. Individuals with early, adult and late onset MS each display distinct features which should be taken in consideration in their treatment.

## 1. Introduction

Multiple sclerosis (MS) is the most common inflammatory autoimmune disease of the central nervous system [1,2]. MS is usually diagnosed between the ages of 20 and 49 years, though in rare cases MS is observed either in childhood and adolescence before the age of 18 years, or at the age of 50 years and later [1]. Age of onset in young adulthood has been considered a major criterion in the diagnosis of MS [3]. However, for two reasons, the importance of age at disease onset has become less important. First, there has been a growing number of cases of persons with MS with an onset during childhood and adolescence or after the age of 50 years. Second, progress in imaging and biomarkers has helped to distinguish MS from mimicry diseases, to identify its pathogenesis and to improve disease management in the longer term [4,5,6].

An onset of MS between the ages of 10 and 18 [7] is termed early onset (EOMS). Prevalence rates range between 2.7% and 10% [7,8,9,10,11,12]. Following Ruet [12], the main indications in individuals with EOMS are high-level inflammatory activities. Alroughani and Boyko [7] similarly observed that, compared to adult MS, early MS is characterized by a higher inflammatory course with frequent infratentorial presentations at onset. Furthermore, individuals with EOMS are at greater risk of developing physical disabilities and cognitive impairments. Next, a clinical challenge is to distinguish EOMS from acute disseminated encephalomyelitis, demyelinating syndromes, autoimmune and systemic pathologies, and infectious, genetic, metabolic and neoplastic diseases. Alroughani and Boyko [7] also stressed that due to the efficient repair mechanisms in childhood and adolescence and compared to adults with MS, early sufferers tend to have much more time to have an Expanded Disability Status Scale (EDSS) score of 6 and higher, though this group reaches an EDSS score of 6 at an earlier point in their lives. 

As regards late onset of MS (LOMS), the consensus is that this be defined as onset at 50 years or older. The prevalence rates of LOMS range from 0.6% to 5% [13] to 12% [14,15]. Noting this spread, Vaughn et al. [16] suggested that the prevalence rates for LOMS have probably appeared to increase as a consequence of increased longevity. Additionally, compared to MS among young adults, the typical characteristic of LOMS are its predominantly progressive course, a greater delay in diagnosis and a higher prevalence of motor disability. The frequency and characteristics of cognitive impairments, on the other hand, appear to be similar to those in young adult onset of MS (AOMS).

To summarize, both EOMS [7,8,9,10,11,12] and LOMS [13,14,15,16] have received considerable attention over the last two decades but there has been some disagreement regarding their clinical characteristics including the course of the disease, initial symptoms, MRI features and disease prognosis. To counter this, the first aim of the present study was to compare the clinical characteristics of those with early onset (EOMS) and with late onset (LOMS) to the characteristics of those with adult onset (AOMS).

A large data set of MS cases ranging from early-onset to late-onset and including evidence collected at several timepoints for each MS case provides an opportunity for a thorough observation of the course of this disease over time, and to identify demographic and clinical factors potentially predictive of its course. Accordingly, the second aim of the present study was to identify any such factors. More particularly, for the EOMS and LOMS cases, we sought to identify factors predicting the transition from a relapsing-remitting MS (RRMS) to a secondary progressive (SPMS).

## 2. Methods 

### 2.1. Study Design and Data Source

In the present study, we used clinical information held in the Isfahan Hakim MS database (IHMSD; Isfahan, Iran). This database was started registering patients with MS and related disorders about seven years ago. We had retrospectively recorded patients’ information until three years ago. From on, information has been recorded prospectively. Data were gathered in the MS clinic, which is a major tertiary referral center affiliated to the Isfahan University of Medical Sciences (Isfahan, Iran). Iran is the second largest country in the Middle East and with a population of approximately 86 million putting it in the top twenty most populous countries in the world. Isfahan is one of the largest provinces, located in the center of Iran with a population of around 5,120,850 habitants in 2016. The contemporary prevalence of MS in Isfahan province is 89/100000 [17]. The clinic covers the greatest number of MS patients in Isfahan and the surrounding province. Routine visits include demographic, clinical and para-clinical information of patients with MS and related disorders. All information of MS patients into a secure and standardized database. All individuals signed a written informed consent. For under-age participants, their legal guardian signed the written informed consent. The regional bioethics committee of the Isfahan University of Medical Sciences (Isfahan, Iran) approved the study, which was performed in accordance with the ethical principles laid down in the current edition (2013) of the Declaration of Helsinki (IR.MUI.MED.REC.1397.222).

### 2.2. Sample

Data of persons with MS (PwMS) attending the MS clinic affiliated to the Isfahan University of Medical Sciences (IUMS; Isfahan, Iran) were entered in the Isfahan Hakim MS database. As mentioned above, the MS clinic in Isfahan covers the catchment area of Isfahan province (about 5.2 million habitants). PwMS attend routine visits at least every six months, or when symptoms worsen, when a PwMS feels unpleasant physiological or psychological changes, or when questions about medications arise. MRI is part of the first assessment and the assessments every two years, or in case that symptoms worsen unexpectedly and dramatically (for further MRI-related information, see below). Following Krupp et al. [18], participants were identified as having early onset MS (EOMS), if at disease onset their age was 18 years or under. Following Lotti et al. [19], participants were identified as late onset of MS (LOMS), if at disease onset they were 50 years or older. Participants were identified as having adult onset MS (AOMS), if at disease onset they were between 18 and 49.9 years old. At the time of diagnosis, all participants fulfilled the revised criteria for MS [4,5,6]. The diagnosis of MS was ascertained by trained and experienced neurologists of the Isfahan MS center. The time between the first manifestations and symptoms of MS and the first visit at the center for the first diagnosis was about two to four days. As mentioned, the time of MS onset was assessed based on a thorough clinical interview, as ascertained by trained and experienced neurologists, and based on criteria extensively described elsewhere [4,5,6]. Further, clinically suspected cases were also tested for aquaporin-4 and anti-myelin oligodendrocyte glycoprotein (MOG) antibody to rule out diseases similar to, but distinct from MS such as neuromyelitis optica spectrum disorder (NMOSD), central nervous system demyelinating disorders or other similar disorders.

### 2.3. Demographic and Clinical Features

Demographic and clinical characteristics were extracted from the database. In addition to age and gender, the following dimensions were extracted:Time of MS onset: time of first manifestations and symptoms of MS.Initial symptoms were categorized into six types: motor dysfunction; sensory disturbances; visual impairments (optic neuropathy); brain stem dysfunction; cerebellar dysfunction; others.Delay in diagnosis: Elapsed time (days) between the first symptoms and the diagnosis of MS.An expert, trained and certified neurologist assessed individuals’ symptom severity using the Expanded Disability Status Scale (EDSS) [20].Following Lublin et al. [21], the course of disease was defined as relapsing-remitting (RRMS), primary progressive (PPMS) or secondary progressive (SPMS).Following others [5,22], a relapse was defined as a neurological deficit lasting for at least 24h; this deficit was associated with an acute or subacute inflammatory event in the central nervous system (CNS), but was not related to fever or infections.Previous studies have shown the predictive role of relapse on disability accumulation using the frequency of relapse in the first two years [23,24]. We therefore extracted the number of attacks in the first two years following disease onset.Relatedly, we also calculated the length of the interval between first and second attack (termed first inter-attack interval).Disease-modifying therapies (DMTs) at follow-up were divided into first-line therapies (interferon beta, glatiramer acetate, dimethyl fumarate, teriflunomide), second- and third-line therapies (fingolimod, natalizumab, rituximab and mitoxantrone).The frequency of autoimmune comorbidity in each patient was evaluated.The history of comorbidity was evaluated at first visit via interview and was further assessed at each subsequent visit.We also obtained the history of MS in first and second-degree relatives of MS patients.The history of other autoimmune disorders in the first-degree relatives was calculated.

### 2.4. Magnetic Resonance Imaging (MRI)

Brain scans with a 1.5 Tesla system (Siemens; Erlangen, Germany) were performed every two years. Radiologists involved were blind to the individuals’ diagnosis, EDSS score, age and other demographic and MS-related information. In the present study, we report data from the MRI taken at the first diagnosis. Diagnosis of MS was obtained according to these MRIs. Further, MRIs were evaluated by trained and experienced neurologists responsible for the diagnosis of MS.

The locations of lesions were classified as follows: supratentorial; infratentorial; whole brain. Spinal cord lesions were classified as cervical and thoracic.

Last, the number of gadolinium-enhancing lesions were reported for the brain and the spinal cord.

MRI is part of the first assessment and the assessments every two years, or in case that symptoms worsen unexpectedly and dramatically. MRIs were performed according to MAGNISM consensus guidelines [25]. The following brain sequences were acquired: Axial; T2W, T2FLAIR, contrast-enhanced-T1W and PD. Sagittal; T2W, STIR, contrast-enhanced-T1W. Further, the following spine sequences were performed: sagittal T2w, STIR, contrast-enhanced T1w. Axial T2w and contrast enhanced T1W.

### 2.5. Statistical Analysis

For metric variables, means and standard deviations are reported. For categorical variables, distributions and frequency are reported.

To calculate mean differences, a series of ANOVAs was performed for the following dimensions: age, age at disease onset, age at follow-up; time lapse between diagnosis and follow-up. Post-hoc analyses (*t*-tests) were performed with Bonferroni–Holm corrections for *p*-values.

To calculate median differences, a series of Kruskal–Wallis-tests was performed for the following dimensions: EDSS score at disease onset; EDSS score at follow-up; first inter-attack-interval; median time for conversion from RRMS to SPMS. Post-hoc analyses (Mann–Whitney *U*-tests) were performed with Bonferroni–Holm corrections for *p*-values.

To calculate differences in distributions, a series of X^2^-tests was performed for the following dimensions: gender, number of autoimmune diseases; occurrence of MS among first- and second-degree family members; family history of autoimmune disorders, current course of disease, prevalent symptoms at disease onset, disease-modifying treatments, location of brain lesions.

Next, we conducted Kaplan–Meier survival curves to explore time to progression (from RRMS to SPMS).

The survival curve differences between EOMS, AOMS and LOMS were tested using log-rank dimensions. A Cox proportional hazards model was carried out to identify the influence of demographic and clinical variables on the time to change from RRMS to SPMS. Predictors were: age at onset, sex (female, male), first manifestation, family history of MS (yes vs. no), autoimmune comorbidity (yes vs. no), EDSS score at diagnosis, number of relapses in the first two years, length of inter-attack interval, type of treatment (first or second-line) spinal lesion (yes vs. no) and gadolinium enhancement (yes vs. no). Due to the limited number of patients who presented brainstem, cerebellar, or other brain lesions at the initial manifestation, these brain lesions were collapsed to a single dimension. If in the univariate model a factor was a statistically significant predictor of the conversion from RRMS to SPMS, this factor was entered as a predictor in the multivariate model.

The level of significance was set at alpha < 0.05. All calculations were performed with SPSS^®^ 24.0 (IBM Corporation, Armonk, NY, USA) for Windows^®^ (Microsoft Corporation, Redmond, WA, USA).

## 3. Results

### 3.1. Sample

A total of 2707 individuals diagnosed with MS were initially identified for in the study. Of these, 80 (2.9%) were excluded due to missing values. The sample with analyzed data therefore consisted of 2627 cases. Of these, 127 (4.8%) were categorized as EOMS, 84 (3.20%) as LOMS and 2416 (91.93%) as AOMS. The mean disease durations (from the onset of symptoms to the last follow-up) were for EOMS 10.10 years (CI:6.1–16.3), for AOMS 7.50 years (CI 4.3–12.17) and for LOMS 5.30 years (CI 2.3–8.5).

Table 1 provides the sociodemographic and basic illness-related information of the three groups (EOMS, AOMS, LOMS).

### 3.2. Early Onset MS

For participants with EOMS (*n* = 127), age at disease onset ranged from 8 to 17 years (mean = 14.89, median = 15, mode = 17). Of these 127, eight (6.2%) were aged 10 or younger at disease onset (termed pre-pubertal MS); 60 (47.2%) were between 11 and 14.9 years old at disease onset; 59 (46.4%) were between 15 and 17.9 years old at disease onset. For the EOMS sub-sample the gender ratio was 3.88:1 (101 females, 26 males). The most prevalent initial symptoms were visual impairments (*n* = 45; 35.4%), brainstem dysfunctions (*n* = 21; 16.5%), sensory disturbances (*n* = 19, 15.0%) and motor dysfunction (*n* = 15, 12.0%). In 31 of the EOMS cases, other autoimmune diseases were present. At the most recent follow-up, 108 EOMS participants (85%) had an EDSS score of ≤3; nine (7.1%) had an EDSS score of 3, seven (5.5%) had an EDSS score of 6; two (1.0%) had an EDSS score of 8; one (0.8%) had an EDSS score higher than 8.

At disease onset, all participants with EOMS were diagnosed with RRMS. At follow-up, 18 participants with EOMS (14.2%) had progressed from RRMS to SPMS. 

### 3.3. Adult Onset MS

For participants with adult onset of MS (AOMS; *n* = 2416; females: *n* = 1921 (79.5%); males: *n* = 495 (20.5%); gender-ratio: 3.88:1) the mean age at onset was 29.42 years (median = 29; mode = 31). The most prevalent initial symptoms were sensory disturbances (*n* = 741, 30.7%), visual impairments (*n* = 632, 26.2%) and motor dysfunction (*n* = 366, 15.1%). At follow-up, 2294 (95%) of the AOMS participants had an EDSS score of 2.9 or lower,142 (5.9%) had an EDSS score of 3, 116 (4.8%) had an EDSS score between 3.1 and 6 and 64 (2.7%) had an EDSS score higher than 6. At disease onset, 2348 (97.2%) of this group were diagnosed with RRMS, while 68 (2.8%) had progressed from RRMS to SPMS. 

### 3.4. Late Onset MS

For participants with late onset of MS (LOMS; *n* = 84; females = 59 (70.2%); males = 25 (29.8%); gender-ratio: 2.36:1), the mean age at onset was 53.97 years (median = 53; mode = 55). At disease onset, seven (8.3%) of this sub-sample were 60 or older. The most prevalent initial symptoms were motor dysfunction (*n* = 32, 38.1%), sensory disturbances (*n* = 26, 31.0%) and visual impairments (*n* = 11, 13.1%).

At follow-up, 73 (87%) had an EDSS score of 2.9 or lower, ten (11.9%) had an EDSS score of 3, nine (10.7%) had an EDSS score between 3.1 and 6 and two (2.4%) had an EDSS score of 8. At disease onset, 42 (74.7% of this sub-sample) were diagnosed with RRMS; 20 (25.3%) were diagnosed as PPMS. At follow-up, 20 (25.3%) had progressed from a RRMS to a SPMS diagnosis.

Figure 1 provides and summarizes the overview of the distribution of age at disease onset, separately for gender and age categories.

Figure 2 provides the overview and summarizes the EDSS scores at disease onset, separately for age categories.

Figure 3 provides the overview and summarizes the distributions of MS categories (RRMS, PPMS, SPMS) at follow-up, and separately for age categories.

### 3.5. Comparisons of Sociodemographic and Illness-Related Features between Participants with Early Onset of MS (EOMS), Adult Onset of MS (AOMS) and Late Onset of MS (LOMS)

Table 1 also provides the statistical comparisons of the sociodemographic and illness-related features between the three groups. Gender distributions did not significantly differ. Age at disease onset, age at follow-up and the time lapse between the first diagnosis and the follow-up assessment did significantly differ; post-hoc analyses with Bonferroni–Holm corrections for *p*-values showed that compared to AOMS and LOMS, plausibly, EOMS were younger at disease onset and at follow-up, and the time lapse from diagnosis to follow-up was longer.

### 3.6. Clinical Characteristics between Participants with Early Onset of MS (EOMS), Adult Onset of MS (AOMS) and Late Onset of MS (LOMS)

Table 2 provides an overview of the clinical characteristics of the three groups.

There were no significant differences in the number of autoimmune comorbidities, the occurrence of MS in the family, the occurrence of family history of other autoimmune disorders, or in the first inter-track interval (in years).

Significant differences were observed as regards the current course of the disease. Compared to participants with AOMS and LOMS, those with EOMS more often had one or two relapses in the first two years. Compared to participants with EOMS and LOMS, those with AOMS more often had zero or 3 ≥ relapses in the first two years. Compared to participants with EOMS and AOMS, those with LOMS more often had zero relapses in the first two years. Figure 4 illustrates graphically this pattern of results.

Significant median differences in EDSS scores were present at first diagnosis and at follow-up. Compared to participants with EOMS and AOMS, those with LOMS had higher EDSS scores at the beginning and at follow-up.

Significant differences were observed in symptoms at onset. Compared to participants with AOMS and LOMS, those with EOMS more often had visual impairments, brain stem symptoms and other symptoms. Compared to participants with EOMS and LOMS, those with AOMS more often had sensory symptoms. Compared to participants with EOMS and AOMS, those with LOMS more often had motor and cerebellar symptoms. 

### 3.7. Results from the MRI; Location of Brain Lesions, Gadolinium-Enhancing Brain Lesions, Spinal Lesions and Gadolinium-Enhancing Spinal Lesions, Separately for Participants with Early Onset of MS (EOMS), Adult Onset of MS (AOMS) and Late Onset of MS (LOMS)

Table 3 provides a statistical overview of the occurrences and locations of brain lesions, gadolinium-enhancing brain lesions, spinal lesions and gadolinium-enhancing spinal lesions, separately for each of the three groups.

There were no significant differences between the three groups in the location of brain lesions (supratentorial, infratentorial, whole brain), in spinal lesions (cervical, thoracic) or in gadolinium-enhancing spinal lesions. Compared to participants with AOMS and LOMS, those with EOMS did have (non-significantly) more gadolinium-enhancing brain lesions.

Further, most patients had both supratentorial and infratentorial lesions which were categorized as patients with whole brain abnormality. The infra-tentorial or supratentorial categories referred to patients who only had either infra-tentorial or supra-tentorial lesion. Given this, the reason for the low number of lesions in these groups is due to the categorization method. Next, most scans performed with 1 cm distance.

### 3.8. Disability and Prognostic Variables between Participants with Early Onset of MS (EOMS), Adult Onset of MS (AOMS) and Late Onset of MS (LOMS)

Figure 5A indicates the median time from RRMS to SPMS, separately for EOMS, AOMS and LOMS, and calculated from disease onset. For EOMS, the median time was 26.4 years (IQR: 21.9–30.8); for AOMS, median time was 20.5 years (IQR: 19.6–21.4); for LOMS, median time was 14.0 years (IQR: 10.7–18.7). The Kruskal–Wallis *H*-test for multiple median comparisons was statistically significant (H > 4.00, *p* < 0.01).

Figure 5B indicates the median age at which RRMS changed to SPMS, separately for EOMS, AOMS and LOMS. For EOMS, the median age was 39.8 years (IQR: 33.7–45.8); for AOMS, the median age was 53 years (IQR: 52.5–54.8); for LOMS, the median age was 67.9 years (IQR: 63.0–72.7). The Kruskal–Wallis H-test for multiple median comparisons was statistically significant (H > 4.00, *p* < 0.01).

Table 4 sets out the calculations for the risk that there is a change from RRMS to SPMS, separately for EOMS, AOMS and LOMS. 

Compared to participants with AOMS, those with EOMS were at higher risk of conversion from RRMS to SPMS (HR: 7.994, 95% CI: 4.833, 13.223).

From disease onset participants with EOMS were at lower risk of conversion from RRMS to SPMS than the LOMS group (HR: 0.463, 95% CI: 0.284, 0.756).

Compared to participants with AOMS, participants with EOMS had a lower risk of conversion from a RRMS to a SPMS (HR: 0.167, 95% CI: 0.101, 0.276). In contrast, from disease onset, and compared to participants with AOMS, those with LOMS were at a higher risk of changing from RRMS to SPMS (HR: 4.226, 95% CI: 2.579, 6.927).

Table 5 provides an overview of the multiple Cox variables regression model to predict conversion from RRMS to SPMS, separately for those with EOMS, AOMS and LOMS.

For participants with EOMS, higher EDSS scores at first visit (HR = 1.287, 95% CI = 10.47, 1.582) predicted progression from RRMS to SPMS.

For participants with AOMS, lower age at disease onset (HR = 1.040, 95% CI=1.021, 1.059), higher EDSS scores at first visit (HR = 1.216, 95% CI = 1.131, 1.309), a shorter first inter-attack interval (HR = 0.947, 95% CI = 0.925, 0.968), and the occurrence of spinal lesions (HR = 1.888, 95% CI = 1.120, 3.182) predicted this progression. In addition, patients with AOMS and with motor impairments were at higher risk of progressing from RRMS to SPMS than those with sensory disturbances (HR = 1.978, 95% CI = 1.319, 2.967).

For participants with LOMS, the occurrence of a spinal lesion (HR = 8.893, 95% CI = 1.702, 46.467) and the presence of spinal gadolinium-enhancement (HR = 32.095, 95% CI=4.075, 252.755) predicted progression from RRMS to SPMS.

## 4. Discussion

The key findings of the present study were as follows. In a large sample of individuals with MS (*N* = 2727), it was possible to distinguish those with early onset (EOMS; *n* = 127), with adult onset (AOMS; *n* = 2416) and with late onset MS (LOMS; *n* = 84), respectively.

For individuals with EOMS, mean age at onset was 14.5 years; key symptoms were visual impairments, brain stem dysfunction, sensory disturbances and motor dysfunctions. On average, 24.6 years after disease onset and at the age of about 39.8 years, 14.2% with RRMS had progressed to SPMS. The key predictor of this change was a higher EDSS score at disease onset. Compared to individuals with AOMS and LOMS, those with EOMS more often had one or two relapses in the first two years, and tended more often to have gadolinium-enhancing brain lesions. 

For individuals with AOMS, mean age at disease onset was 29.4 years; key symptoms were sensory disturbances, motor dysfunctions and visual impairments. On average, 20.5 years after disease onset, and thus when on average they were 53, 15.6% with RRMS switched to SPMS. The key predictors of this change were, at disease onset, a higher EDSS score, lower age, a shorter inter-attack interval and spinal lesions. Compared to individuals with EOMS and LOMS, individuals with AOMS more often had either no or three or more relapses in the first two years.

For individuals with LOMS, mean age at onset was 53.8 years; key symptoms were motor dysfunctions, sensory disturbances and visual impairments. On average, 14 years after disease onset and at the age of about 67 years, 25.3% with RRMS had progressed to SPMS. The key predictors were, at disease onset, occurrence of spinal lesions and spinal gadolinium-enhancement. Compared to individuals with EOMS and AOMS, those with LOMS more often had no relapses in the first two years; they also had higher EDSS scores at the disease onset and at follow-up. 

Here, we discuss some specific results.

As regards progression from RRMS to SPMS, the present results are comparable to those reported elsewhere [26,27,28,29]. However, in individuals with EOMS, the change from RRMS to SPMS occurred at a younger age than those with AOMS or LOMS. It follows that individuals with EOMS might face a more negative disease course in the longer term. To gain more insight in the disease progress in individuals with EOMS, much longer-term studies would be helpful. Furthermore, as noted above, for those with EOMS, AOMS and LOMS, respectively different factors predicted disease progression

As regards the course of disease for MS sufferers, the proportion with primary and secondary progressive courses (PPMS; SPMS) increased with advancing age. Not surprisingly, in the present study no individual with EOMS had a primary progressive course. This is consistent with previous studies [26,30,31] which have shown younger age to be associated with the course of MS. This observation has been explained by reference to the longer-lasting degeneration and smaller inflammatory processes typically observed among older individuals that are related to the progressive phases of MS [32,33,34]. Similarly, greater age has been negatively associated with lower immune cell activity, along with a lower expression of the gene and oligodendrocyte progenitor recruitment and differentiation [35,36]. It follows that the capacity for repair, remyelination and other functions required to stabilize a relapsing-remitting course appeared to decline [37]. The present findings add to and confirm the substantial impact of age-related physiological changes on the course of the disease and accumulation of disabilities. 

As regards the pattern of symptoms at disease onset, the present results match those reported in previous studies [10,26,38,39]. As noted above, for individuals with EOMS, the key symptoms were visual impairments, brain stem dysfunction, sensory disturbances and motor dysfunctions. For individuals with AOMS, the key symptoms were sensory disturbances, motor dysfunctions and visual impairments. For individuals with LOMS, the key symptoms were motor dysfunctions, sensory disturbances and visual impairments. Thus, we conclude that the present pattern of results could be helpful in supporting diagnostic algorithms. 

As regards Expanded Disability Status Scale (EDSS) scores at disease onset, several studies have indicated that a higher EDSS score could be predictive of the course of the disease [40,41]. In the present study, we found EDSS scores at the time of the diagnosis among individuals with EOMS and AOMS to be similar, while individuals with LOMS had significantly higher EDSS scores. Furthermore, in all groups, higher EDSS scores at the first visit was associated with an increased risk of a negative progression of the disability. However, and in contrast, in individuals with LOMS, the multivariate analysis showed that the EDSS score was not an independent predictor of the conversion from RRMS to SPMS course. 

As regards the frequency of relapses in the first two years, individuals with EOMS had more relapses within the first two years than those with either AOMS or LOMS; this pattern accords with previous findings [42]. Additionally, among individuals with AOMS and in agreement with Scalfari et al. [24], we observed that a shorter first-attack interval predicted an earlier conversion from RRMS to SPMS. This pattern, however, was not observed among individuals with EOMS or LOMS.

As regards the prevalence rates of MS among family members, there were no significant differences between the groups (EOMS, AOMS, LOMS) for the frequency of MS among first and second-degree relatives. Likewise, the rate of autoimmune diseases in first-degree relatives did not vary between the groups. In addition, although the rate of autoimmune comorbidity was higher in individuals with EOMS than in those with AOMS and LOMS, the differences were not significant. This pattern of results is consistent with some previous findings [43] but not with others reporting a younger age at MS onset to be associated with a higher risk of MS among siblings [44,45]. 

As regards the MRI results, differences between the groups in brain locations were very small; this reflects previous studies [9,38,46]. The proportion of individuals with a gadolinium-enhancing lesion at diagnosis date was slightly lower at greater age; this fits with previous findings [38,47], as contrast enhancement has been related to blood-brain barrier disruptions and to an inflammation phase of disease and lesion [38]. It follows that greater age may be associated with a decline in inflammation activities, which in turn may be responsible for the lower frequency of gadolinium enhancement in older patients.

Next, according to the McDonald criteria, spinal lesion is considered as one of the four topographic points that are required to demonstrate dissemination in space [5]. However, few studies have determined the frequency of spinal lesions in early and late onset MS. We found that 54% of patients had spinal lesions and this rate matches previous studies of individuals with EOMS [48]. For individuals with LOMS, spinal lesion rates were within the range 36%–81% [38,49], and our prevalence rate of 57% falls within this range. Next, while spinal lesions and gadolinium-enhancement were the only predictors of disability progression in LOMS, this needs to be replicated with a larger sample. Nonetheless, our findings highlight the predictive role of spinal lesions in MS sufferers. We conclude that adults with MS (AMOS; LOMS) and with manifestations related to spinal abnormalities should be assessed through imaging; likewise, spinal lesions should be considered when it comes to treatment and management of the disease.

Despite the novelty of the study, various limitations warrant against overgeneralization of the findings. First, the study is retrospective with the risk of some missing information. Second, the sub-samples of EOMS and LOMS cases were small, though we believe this reflects their actual prevalence rates. Third, almost by definition, the follow-up period for individuals with LOMS was short. Fourth, while data from MRIs are helpful in supporting diagnosis, more fine-grained images would have been helpful to localize brain lesions in a much more precise fashion. Fifth, psychological dimensions were not assessed; in particular, it was not possible to relate the sociodemographic and physiological data to psychological dimensions such as quality of life, sleep quality [50,51,52,53,54], regular physical activity [53,55,56,57], cognitive performance [58] or social cognition and empathy [53].

Next, although not the focus of the present study, a deepened and thorough comparison of the present data from Iran with other large published natural history cohorts from Western countries such as the data set from Lyon [59] or from Vancouver [60] have allowed an even more detailed discussion of the pattern of results observed in Iran. Roughly, a total number of 1844 of individuals with MS were enrolled in Lyon cohort study [59]. In accordance with our study, the median time for transition from relapsing to secondary course was 19 years. However, the time of transition to a progressive phase had a range of 5 to 25 years in the cohort study from Lyon [59]. The results from this cohort study suggested that some factors including age at onset, first inter-attack interval number of attacks in the first two or five years after disease onset, and spinal lesion were associated with the time of conversion from a relapsing to a progressive phase. Further, a retrospective record-linkage cohort study was published on 2841 individuals with MS from British Columbia Multiple Sclerosis database [60]. While in the present study we evaluated only clinic cases, the British Columbia Multiple Sclerosis database study reported both clinic and non-clinic cases. It turned out that in the latter study non-clinic cases had older age, suffered from more comorbidities, and that a limited number of them had filled a prescription for a DMT. Thus, from the British Columbia Multiple Sclerosis database study [60] we learned that non-clinic individuals with MS should be thoroughly integrated and followed-up in all registrations.

## 5. Conclusions

In a large sample of individuals with MS, cases of early onset (EOMS) and late onset of MS (LOMS) are less frequent though, compared to individuals with adult onset of MS (AOMS), those with EOMS and LOMS displayed distinctive features which should be taken into consideration in their treatment. 

## Figures and Tables

**Figure 1 jcm-09-01326-f001:**
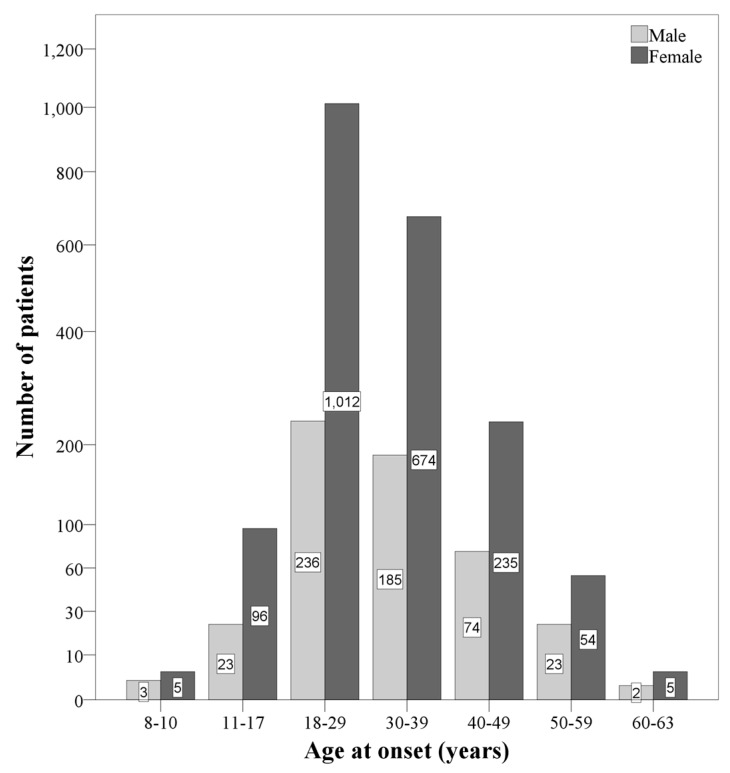
Overview of the distribution of participants, separately for gender and age categories. The female/male-ratios within the age categories are as follows: 8–10 years: 1.66:1; 11-17 years: 4.17:1; 18–29 years: 4.28:1; 30–39 years: 3.64:1; 40–49 years: 3.17:1; 50–59 years: 2.34:1; 60–63 years: 2.5:1.

**Figure 2 jcm-09-01326-f002:**
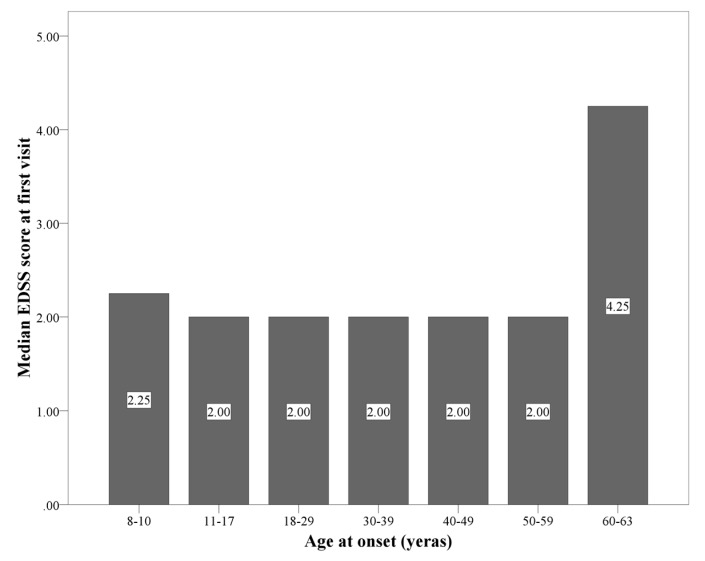
The median of Expanded Disability Status Scale (EDSS) score (IQR) within the age categories are as follows: 8–10 years: 2.25 (0.5, 3.0); 11–17 years: 2.0 (0.0, 2.0); 18–29 years: 2.0 (1.0, 2.0); 30–39 years: 2.0 (1.0, 2.0); 40–49 years: 2.0 (1.5, 3.0); 50–59 years: 2.0 (2.0, 3.0); 60–63 years: 4.25 (2.75, 5.625).

**Figure 3 jcm-09-01326-f003:**
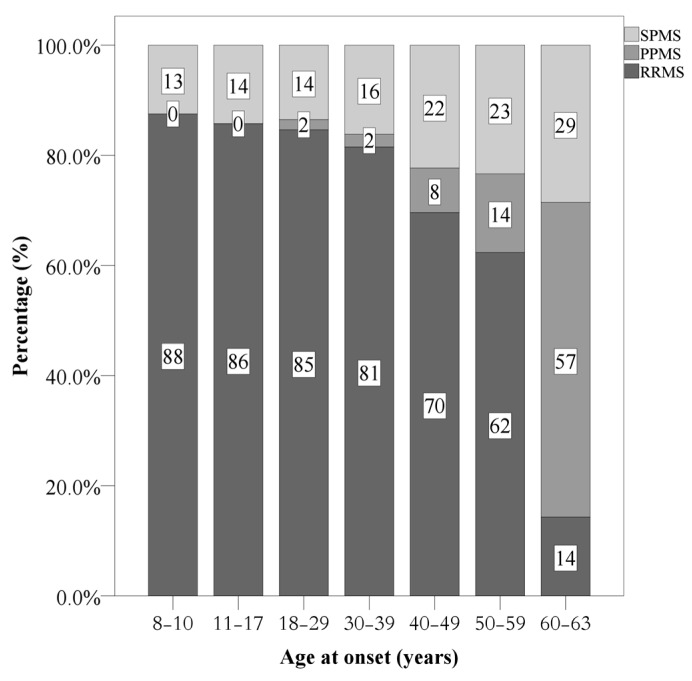
Frequency of disease courses at the last visit, separately for age categories at disease onset. RRMS = relapsing remitting MS. PPMS = primary progressive MS; SPMS = secondary progressive MS.

**Figure 4 jcm-09-01326-f004:**
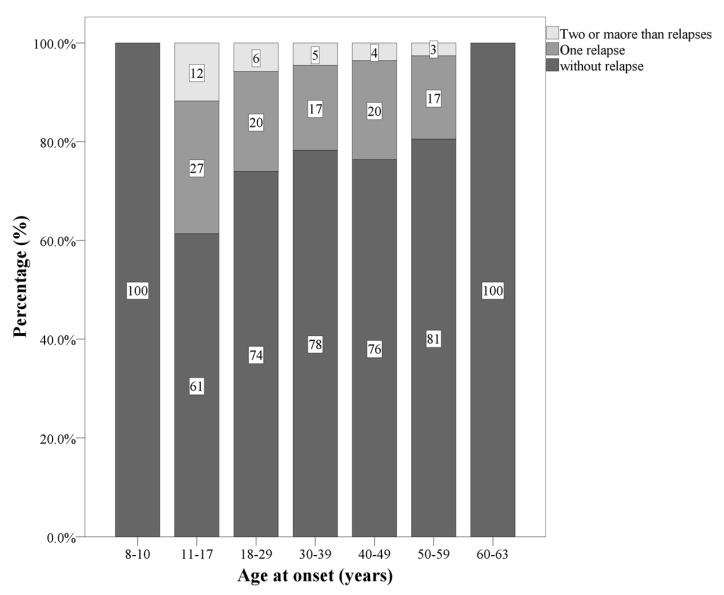
Number of relapses, separately for age ranges.

**Figure 5 jcm-09-01326-f005:**
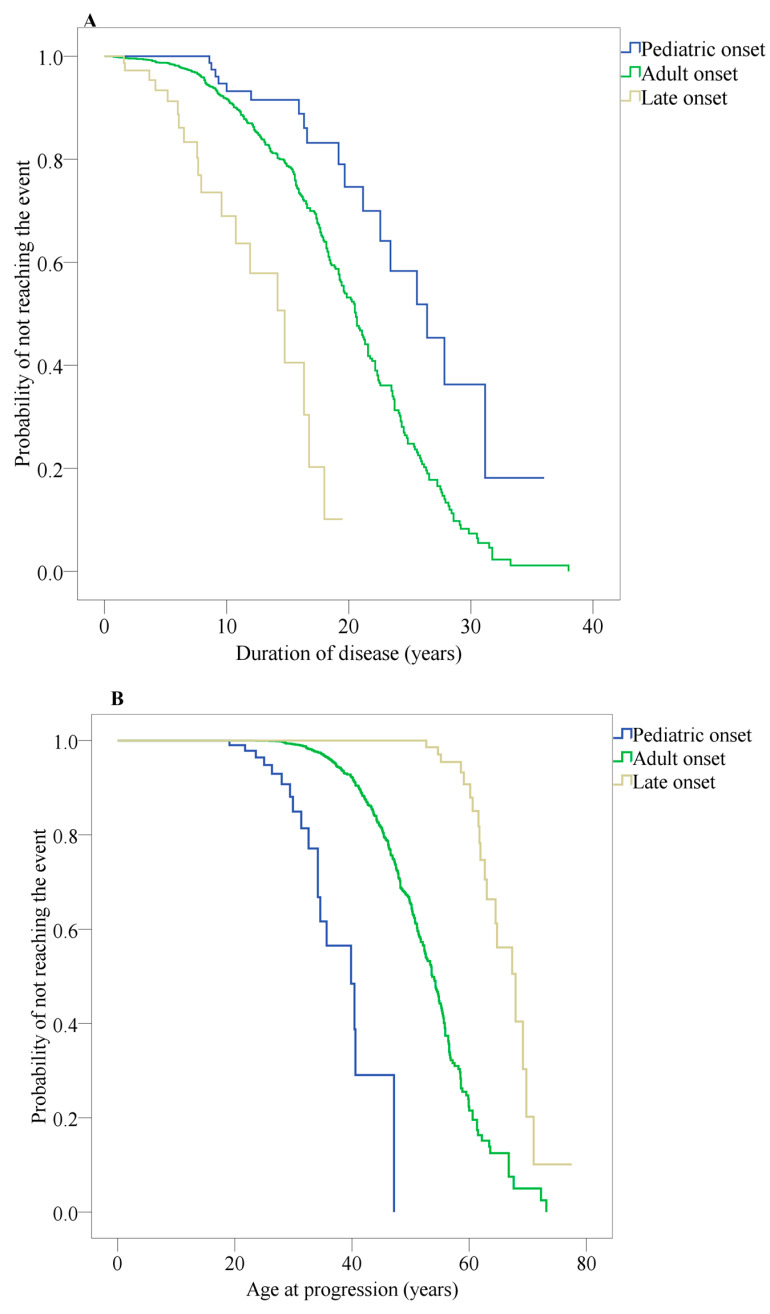
Kaplan–Meier survival curves of the probability estimates of not reaching secondary progression. (**A**) Time from disease onset to conversion from relapsing remitting MS (RRMS) to a secondary progressive MS (SPMS). (**B**) Time from birth to conversion from relapsing remitting MS (RRMS) to a secondary progressive MS (SPMS).

**Table 1 jcm-09-01326-t001:** Descriptive and inferential statistical indices of demographic characteristics between participants with early onset multiple sclerosis (EOMS), adult onset (AOMS) and late onset (LOMS).

		Samples			Statistics (between Groups)	
	Whole sample	Early onset of MS (EOMS)	Adult onset of MS (AOMS)	Late onset of MS (LOMS)		Post-hoc analysis
N	2627	127	2416	84		
	*n* (%)	*n* (%)	*n* (%)	*n* (%)		
Sex (females)	2081 (79.2%)	101 (79.5%)	1921 (79.5%)	59 (70.2%)	X^2^(N = 2627, df = 2) = 4.25	-
	M (SD)	M (SD)	M (SD)	M (SD)		
Current age (years)	37.6 (9.87)	25.20 (7.15)	37.42 (8.85)	58.79 (5.47)	F(2, 2624) = 378.85 *** pε^2^ = 0.224 (L)	EOMS < AOMS EOMS < LOMS AOMS < LOMS
Age at onset (years)	30.21 (9.04)	14.8 (2.29)	30.00 (7.50)	53.97 (3.35)	F(2, 2624) = 336.47 *** pε^2^ = 0.360 (L)	EOMS < AOMS EOMS < LOMS AOMS < LOMS
Time lapse from diagnosis to follow-up	8.64 (6.06)	11.47 (7.34)	8.59 (5.99)	5.93 (4.57)	F(2, 2624) = 22.66 *** pε^2^ = 0.017 (T)	EOMS > AOMS EOMS > LOMS AOMS > LOMS

Notes: *** = *p* < 0.001; T = trivial effect size; L = large effect size; ≥statistically significantly larger than; ≤statistically smaller than. MS = multiple sclerosis, SD = standard deviation.

**Table 2 jcm-09-01326-t002:** Descriptive and inferential statistical indices of clinical characteristics between participants with early onset (EOMS), adult onset (AOMS) and late onset Multiple Sclerosis (LOMS).

			Samples			Statistics (between Groups)	
		Whole sample	Early onset of MS (EOMS)	Adult onset of MS (AOMS)	Late onset of MS (LOMS)		Post-hoc tests
*N*		2627	127	2416	84		
		*n* (%)	*n* (%)	*n* (%)	*n* (%)		
Autoimmune comorbidity (yes vs. no)		440 (16.7%)	31 (24.4%)	402 (16.6%)	7 (8.3%)	X^2^(*N* = 440, df = 2) = 3.42	
MS familial history	First degree (yes vs. no)	250 (9.5%)	15 (11.8%)	227 (9.4%)	8 (9.5%)	X^2^(*N* = 632, df = 3) = 2.45	
	Second degree (yes vs. no)	382 (14.5%)	14 (11.0%)	362 (15.0%)	6 (7.1%)		
Family history of other autoimmune disorder (first degree) (yes vs. no)		125 (4.8%)	4 (3.1%)	115 (4.8%)	6 (7.1%)	X^2^(*N* = 125, df = 1) = 1.95	
Current disease course	Relapsing remitting MS	2129 (81.0%)	109 (85.8%)	1971 (81.6%)	49 (58.3%)	X^2^(*N* = 2627, df = 4) = 71.83 ***	
	Secondary progressive MS	415 (15.8%)	18 (14.2%)	377 (15.6)	20 (23.8%)		POMS: more RRMS, less SPMS, low PPMS AOMS: more LOMS: more sPMS and PPMS
	Primary progressive MS	83 (3.2%)	-	68 (2.8%)	15 (17.9%)		
Number of patients with relapse in the first two years	0/1/2/3≥	1981/508/122/16	81/32/14/0	1831/463/106/16	69/13/2/0	X^2^(*N* = 2627, df = 6) = 19.57 ***	POMS: 1 and 2 relapses ↑ AOMS: zero and 3≥ relapsaes ↑ LOMS: 0 relapses ↑
		Median (range)	Median (range)	Median (range)	Median (range)		
EDSS at first visit		2.0 (1.0–2.5)	2.0 (0.0–2.0)	2.0 (1.0–2.5)	2.25 (2.0–3.0)	H(*N* = 2608, df = 2) = 31.38 ***	LOMS > POMS; AOMS
EDSS at last follow-up		1.0 (0.0–2.5)	0.0 (0.0–2.0)	1.0 (0.0–2.5)	2.25 (0.75–5.0)	H(*N* = 2608, df = 2) = 27.69 ***	LOMS > POMS; AOMS
First inter-attack interval (years)		3.0 (1.0–5.0)	2.0 (1.0–4.0)	3.0 (1.0–5.0)	2.0 (1.0–5.0)	H(*N* = 2608, df = 2) = 4.54	
		*n* (%)	*n* (%)	*n* (%)	*n* (%)		
Prevalent symptoms at onset	Sensory	786 (29.9%)	19 (15.0%)	741 (30.7%)	26 (31.0%)	X^2^(*N* = 2627, df = 10) = 70.75 ***	
	Visual	688 (26.2%)	45 (35.4%)	632 (26.2%)	11 (13.1%)		
	Motor	417 (15.9%)	19 (15.0%)	366 (15.1%)	32 (38.1%)		POMS: visual ↑; brain stem ↑; other ↑
	Brainstem	299 (11.4%)	21 (16.5%)	277 (11.5%)	1 (1.2%)		AOM: sensory ↑; LOMS: motor ↑ celebellar ↑; visual ↓
	Cerebellar	161 (6.1%)	3 (2.4%)	148 (6.1%)	10 (11.9%)		
	Others	97 (3.7%)	6 (4.7%)	90 (3.7%)	1 (1.2%)		
Disease modifying treatments at the last visit	First line	1709 (65.1%)	63 (49.6%)	1585 (65.6%)	61 (72.6%)	X^2^(*N* = 2627, df = 4) = 28.14 ***	POMS: first ↓; second ↑; switch ↑
	Second/third line	918 (34.9%)	64 (50.4%)	831 (34.4%)	23 (27.4%)		AOMS: first ↑
	switch						LOMS first ↑; second ↓

Notes: *** = *p* < 0.001. ↑ = higher than; ↓ = lower than.

**Table 3 jcm-09-01326-t003:** Descriptive and inferential statistical indices of MRI features between participants with early onset (EOMS), adult onset (AOMS) and late onset multiple sclerosis (LOMS).

			Samples			Statistics (between Groups)
		All patients *N* = 2627	EOMS *N* = 127	AOMS *N* = 2416	LOMS *N* = 84	
		*n* (%)	*n* (%)	*n* (%)	*n* (%)	
Location of brain lesion	Supratentorial	725 (27.6%)	24 (18.9%)	674 (27.9%)	21 (25.0%)	X^2^ < 1.5, *p* = 0.115
	Infratentorial	9 (0.4%)	-	10 (0.4%)	1 (1.2%)	
	Whole brain	1891 (72.0%)	103 (78.7%)	1732 (71.7%)	62 (73.8%)	
Gadolinium -enhancing brain lesion ≥ 1		333 (12.6%)	22 (17.3%)	306 (12.7%)	5 (6.0%)	X^2^(N = 333, df = 2) = 5.43 *
Spinal lesion ≥ 1	Cervical	1557 (59.3%)	69 (54.3%)	1441 (59.6%)	47 (56.0%)	X^2^ < 1.5, *p* = 0.405
	Thoracic	87 (3.3%)	4 (3.1%)	80 (3.3%)	3 (3.6%)	X^2^ < 1.0, *p* = 0.986
	Total	1582 (60.2%)	69 (54.3%)	1465 (60.6%)	48 (57.1%)	X^2^ < 1.5, *p* = 0.309
Gadolinium -enhancing spinal lesion ≥ 1		94 (3.6%)	8 (6.3%)	119 (5.0%)	2 (2.4%)	X^2^ < 1.5, *p* = 0.432

EOMS = early onset multiple sclerosis; AOMS = adult onset multiple sclerosis; LOMS = late onset multiple sclerosis. * = *p* < 0.05.

**Table 4 jcm-09-01326-t004:** Univariate cox regression of variables association with time to conversion to the state of secondary progressive MS (SPMS).

Variable		Early Onset MS		Adult Onset MS		Late Onset MS	
		HR (95% CI)	*p*-Value	HR (95% CI)	*p*-Value	HR (95% CI)	*p*-Value
Age at onset		1.139 (0.937, 1.386)	0.191	1.054 (1.040, 10.68)	**0.000**	1.145 (0.993, 1.320)	0.063
Sex (reference = Female)		1.745 (0.600, 5.073)	0.307	1.253 (0.989, 1.588)	0.062	1.211 (0.383, 3.828)	0.744
MS familial (reference = No)		0.648 (0.184, 2.283)	0.499	0.917 (0.663, 1.269)	0.601	2.067 (0.457, 9.353)	0.346
Autoimmune comorbidity (reference = No)		0.043 (0.00, 101.850)	0.427	0.815 (0.519, 1.280)	0.374	1.366 (0.308, 6.065)	0.682
Symptom of onset (reference = Sensory)	Visual	1.234 (0.237, 6.428)	0.803	0.686 (0.492, 0.959)	0.026	0.689 (0.110, 4.304)	0.691
	Motor	4.409 (0.677, 28.705)	0.121	1.952 (1.445, 2.637)	**0.000**	2.301 (0.601, 8.808)	0.224
	Other	3.016 (0.577, 15.777)	0.191	1.223 (0.909, 1.645)	0.184	1.975 (0.467, 8.361)	0.355
EDSS at first visit		1.287 (10.47, 1.582)	**0.016**	1.209 (1.155, 1.265)	**0.000**	1.518 (1.35, 2.030)	**0.005**
Attack in first two years (reference = No)		0.895 (0.284, 2.814)	0.849	1.124 (0.887, 1.425)	0.333	0.919 (0.203, 4.171)	0.913
Length of first attack interval		0.953 (0.848, 1.071)	0.419	0.961 (0.942, 0.980)	**0.000**	0.939 (0.754, 1.170)	0.577
Brain gadolinium -enhancement (reference = not applied)		2.118 (0.558, 8.034)	0.270	0.456 (0.293, 0.711)	**0.001**	0.044 (0.000, 362.441)	0.497
Spinal MRI (reference = normal)		1.945 (0.726, 5.209)	0.185	1.722 (1.354, 2.191)	**0.000**	1.787 (1.341, 10.697)	**0.012**
Spinal gadolinium-enhancement (reference = not applied)		2.044 (0.261, 16.034)	0.496	0.582 (0.327, 1.035)	0.065	4.642 (1.018, 21.163)	**0.047**

Notes: HR = hazard ratio; CI = confidence interval; Ref = reference; EDSS = Expanded Disability Status Scale, Gd: gadolinium.

**Table 5 jcm-09-01326-t005:** Multivariate cox regression model to predict secondary progressive MS conversion in the groups.

Variable		HR (95% CI)	95% CI
Early-onset MS			
EDSS at first visit		1.29 (10.47–15.82)	0.016
Adult-onset MS			
Age at onset		1.04 (1.02–1.06)	0.000
EDSS at first visit		1.22 (1.13–1.31)	0.000
Length of first attack interval		0.95 (0.93–0.97)	0.000
Spinal gadolinium-enhancement (reference = not applied)		0.86 (0.52–1.43)	0.557
Spinal MRI (reference = normal)		1.89 (1.12–3.18)	0.017
Symptom of onset (Ref. = Sensory)	Visual	0.82 (0.54–1.25)	0.347
	Motor	1.98 (1.32–2.97)	0.001
	Other	0.99 (0.66–1.51)	0.990
Late-onset MS			
EDSS at first visit		1.35 (0.97–1.88)	0.073
Spinal MRI (reference = normal)		8.89 (1.70–46.47)	0.010
Spinal gadolinium-enhancement (reference = not applied)		32.09 (4.08–252.76)	0.001

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
