# Peer review of "Clinical Characteristics and Disability Progression of Early- and Late-Onset Multiple Sclerosis Compared to Adult-Onset Multiple Sclerosis"

_jcm, 2020, doi:10.3390/jcm9051326_

Round 1

Reviewer 1 Report

In this study clinical/demographic/radiological features are compared among MS patients with early, adult and late age onset. The study is advantaged by a large sample size, a geographically based methodology with low ascertainment bias, and sound statistical methodology. I have no major concerns and comments, but I would suggest: 

  • to revise the manuscript grammar and style, as it does not read smooth in many sessions
  • to consider shortening the manuscript and reducing the number of tables (eg table 4, 5 and 6 provide similar information)
  • Figure 4 is missing from the manuscript
  • The conclusions should be tightened in order to avoid unnecessary repetition of results.   

Author Response

We thank Reviewer 1 for her/his valuable comments. We answered to her/his comments with much more details in the point-by-point-response. 

Reviewer 2 Report

Mirmosayyeb et al examined clinical characteristics and disease progression from RRMS to SPMS in different cohorts of MS patients, stratified for age at onset of disease. Design of the study was a review of registry data from a large MS database. The research question is important and appropriate methodology was used.

I have some points to consider:

  1. Methods, line 104: This is not a prospective study, unless the specific study outline and research questions were already determined at start of the registry. See also line 439.
  2. Methods: Details of the registry are missing. Are patient visits done at a regular basis, is a specific protocol for clinical / ancillary examinations (MRI, etc.) followed, are only incident cases captured, when was the registry started, how many study centers, etc.
  3. Methods, line 132: how was time of MS onset assessed? from medical history? How long was the delay between time of MS onset and first visit at the study center? Who confirmed diagnosis of MS? Only study centers?
  4. Methods, line 157: was a routine MRI program done? Which sequences were done?
  5. Methods, line 163: please add “number of” Gadolinium-enhancing lesions …
  6. Figure 2: since EDSS is a categorical parameter, I recommend to show median + IQR
  7. Table 2: please specify if second EDSS is “EDSS at LAST follow-up”
  8. Table 2: please specify if DMTs during whole observation period or at last visit are shown
  9. Table 2: since there is a significant difference in the number of relapses in the first 2 years between the 3 groups and the data is difficult to interpret in the table (without seeing frequencies), I recommend visualization, similar to Fig.3
  10. Results, line 286: when was the MRI done? at disease onset, confirmation of diagnosis, or later? Were actually MRIs evaluated or radiologic reports?
  11. Table 3: in all 3 groups there is a surprising low number of supratentorial and infratentorial lesions. Please give the percentage of scans for which detailed data on lesion distribution were available.
  12. Figure 4A and B are missing.
  13. Discussion: It could be worthwhile to compare the Iranian data with other large published natural history cohorts, e.g. from Lyon or Vancouver.

Author Response

We thank Reviewer 2 for her/his valuable comments. We answered to her/his comments with much more details in the point-by-point-response. 

Round 2

Reviewer 2 Report

Thank you for the revision. Most of my points were solved.

Comment 6/Figure 2: Hard to believe that this figure now should display median values, since these are exactly same to the previous version showing mean values. Also the 25/75 IQR (interquartile range) is missing.

Comment 13: in my opinion, this should not be a limitation, but a task that easily could be completed

Figure 4: thank you for including this data as Figure. To better explain why all patients in the age group 8-10 yrs are without relapses, I recommend to show the number of patients included and the duration of observation. I guess both is low in this age group. 

Author Response

Again, we thank Reviewer #2 for her/his further comments and scrutiny. We have addressed her/his concerns. Please find the detailed comments in the point-by-point-response attached as a separate file.